# Unlocking the Potential of Molecularly Imprinted Polydopamine in Sensing Applications

**DOI:** 10.3390/polym15183712

**Published:** 2023-09-09

**Authors:** Abderrahman Lamaoui, Abdellatif Ait Lahcen, Aziz Amine

**Affiliations:** 1Process Engineering and Environment Lab, Chemical Analysis & Biosensors Group, Faculty of Science and Techniques, Hassan II University of Casablanca, B.P. 146, Mohammedia 28806, Morocco; 2Weill Cornell Medicine, Cornell University, New York, NY 10021, USA; abdo.aitlahcen@gmail.com

**Keywords:** molecularly imprinted polymer, polydopamine, sensors, self-polymerization

## Abstract

Molecularly imprinted polymers (MIPs) are synthetic receptors that mimic the specificity of biological antibody–antigen interactions. By using a “lock and key” process, MIPs selectively bind to target molecules that were used as templates during polymerization. While MIPs are typically prepared using conventional monomers, such as methacrylic acid and acrylamide, contemporary advancements have pivoted towards the functional potential of dopamine as a novel monomer. The overreaching goal of the proposed review is to fully unlock the potential of molecularly imprinted polydopamine (MIPda) within the realm of cutting-edge sensing applications. This review embarks by shedding light on the intricate tapestry of materials harnessed in the meticulous crafting of MIPda, endowing them with tailored properties. Moreover, we will cover the diverse sensing applications of MIPda, including its use in the detection of ions, small molecules, epitopes, proteins, viruses, and bacteria. In addition, the main synthesis methods of MIPda, including self-polymerization and electropolymerization, will be thoroughly examined. Finally, we will examine the challenges and drawbacks associated with this research field, as well as the prospects for future developments. In its entirety, this review stands as a resolute guiding compass, illuminating the path for researchers and connoisseurs alike.

## 1. Introduction

It is widely acknowledged that, while every individual possesses their own distinct set of fingerprints, each finger exhibits its unique pattern as well. This analogy drawn from the realm of fingerprints finds relevance in the context of chemical species, where each molecule bears its distinctive imprint characterized by specific size, functional group, and shape attributes. To generate such molecular imprints, a polymerization process is employed, wherein the target molecule is incorporated within the polymer structure. The subsequent elimination of the target molecule yields a polymer endowed with specific imprints tailored for the desired analyte. This resulting polymer is referred to as a Molecularly Imprinted Polymer (MIP) [1,2].

MIPs are synthetic receptors that emulate the affinity seen in biological antibody–antigen interactions. These polymers function through a “lock and key” mechanism, selectively capturing the molecules they were templated with during the polymerization process [3]. When contrasted with natural receptors, MIPs present a range of benefits. These encompass enhanced durability, economical production, and strong resilience against both mechanical stresses and chemical influences [4,5,6]. Moreover, MIPs have found widespread applications in various fields such as biosensors [7], sensors [8,9], paper-based analytical devices [10], cancer treatment [11], solid-phase extraction (SPE) [12,13], competitive assays [14], and chromatography [15].

Although the polymerization method is a critical phase, the selection of monomers is a crucial aspect in designing MIPs with desired recognition properties. The most used functional monomers are methacrylic acid [16], acrylamide [17], vinyl pyridine [18], acrylic acid [19], and pyrrole [8]. Several factors are taken into consideration when choosing monomers for MIP synthesis such as functional groups of monomers, functional groups of templates, the polymerization method, etc.

Dopamine, a small molecule, exhibits susceptibility to undergo polymerization under mildly alkaline conditions through oxidation mechanisms. This polymerization process gives rise to a cohesive and adherent material known as polydopamine (PDA), which draws inspiration from the adhesive properties observed in mussel adhesive proteins [20,21,22]. Due to its exceptional adhesive properties, simplicity, versatility, and broad applicability across various fields such as biomedical, energy, and microfluidics, PDA derived from mussel-inspired strategies has emerged as a highly potent polymer for surface modification [23,24]. In 2007, Ouyand et al. reported for the first time the use of dopamine as a monomer for the preparation of MIP [25]. The MIPda was prepared electrochemically for the enantioselective recognition of glutamic acid. One year later, the same group reported the chemical synthesis of MIPda for protein recognition [26]. The mussel-inspired molecularly imprinted polymer (m-MIP) possesses several advantageous attributes, including notable hydrophilicity, excellent biocompatibility, and the ability to control thickness. These qualities collectively render it an appealing recognition element for sensor applications. Since its initial introduction, the utilization of dopamine as a monomer in MIP synthesis has gained extensive popularity [27,28]. MIPda has found diverse applications, ranging from environmental [29] and food [30] monitoring to clinical diagnostics [31,32], thereby showcasing their broad utility.

Given the current surge in research activities focusing on MIPs based on PDA [33], we recognize the pressing need to publish a comprehensive review that not only establishes and discusses recent applications but also identifies challenges and paves the way for future investigations. This proposed review aims to shed light on the extensive utilization of PDA in MIPs. It will offer a thorough overview encompassing diverse aspects, including the various support materials employed for the preparation of MIPda. In addition, the different synthesis methods of MIPda and its characteristics were deeply discussed. Furthermore, it will delve into the broad range of sensing applications demonstrated by MIPda, encompassing the detection of ions, small molecules, epitopes, proteins, viruses, and bacteria. Additionally, we will address the challenges and limitations inherent in this field of research, while exploring the prospects for future advancements. By unlocking the full potential of MIPda, we aspire to ignite further research and development endeavors within this highly promising domain.

## 2. Core/Substrate Materials Used to Be Modified with Molecularly Imprinted Polydopamine (MIPda)

Different core/substrate materials are listed, which have been modified with MIPda for specific templates (Table 1). These materials include magnetic nanoparticles, metal/metal oxide substrates, carbonaceous materials, quantum dots, silica nanoparticles, and QCM crystals.

The core/substrate materials used in the modification of MIPda play crucial roles in determining the properties and functionalities of the resulting materials. Magnetic nanoparticles, for example, not only provide magnetic functionalities but also exhibit nanozyme properties, enabling catalytic activities at the nanoscale (Figure 1A) [34]. Magnetic nanoparticles are not stable over time and can be destroyed during the template removal when using an acidic medium. Therefore, magnetic nanoparticles are usually pre-modified with SiO_2_ [35], fibrous SiO_2_ [36], and polymer [37]. The presence of the SiO_2_ shell serves a crucial function by safeguarding the Fe_3_O_4_ NPs against oxidation. Moreover, it offers significant stability and adaptability for surface modifications [38]. The SiO_2_ shell is compact and does not have a high specific area leading to MIPda with low adsorption capacity. Chen et al. addressed this issue by modifying Fe_3_O_4_ NPs with fibrous SiO_2_ shells [36]. The modification of Fe_3_O_4_ with polymer, for example, Polymethyl methacrylate (PMMA) was developed to effectively immobilize the template protein “lysozyme” on the surface of Fe_3_O_4_/PMMA before the growth of PDA. However, PMMA is not a good support to immobilize lysozyme to prepare a protein-imprinted polymer [37].

Metal/metal oxide substrates contribute to the development of MIPda-based materials with enhanced photoelectrochemical properties, allowing for efficient energy conversion and sensing applications [39]. Moreover, these substrates facilitate improved electron transfer kinetics, leading to enhanced sensitivity and response in the presence of target analytes (Figure 1B,C) [40,41,42]. Additionally, they offer increased affinity for proteins, making them suitable for selective recognition and binding [43]. Carbonaceous materials, such as carbon nanotubes (Figure 1D) [44], reduced graphene oxide [45], carbon fibers [32], and pencil graphite electrodes [46] not only impart conductive properties [44] to the MIPda-based materials but also enable surface modification, allowing for the incorporation of additional functional groups or nanoparticles [32]. Quantum dots, on the other hand, provide superb optical properties to MIPda-based materials, including high fluorescence and tunable emission, making them ideal for optical sensing and imaging applications [47]. Their monodispersed surface characteristics further enhance their performance, ensuring uniform and reproducible results. Silica nanoparticles were used as supporting matrices to prepare MIPda and enable a well-adhered MIPda (Figure 1E) [48]. Lastly, QCM crystals serve as excellent supports for quartz crystal microbalance, enabling the real-time monitoring of molecular interactions and facilitating protein recognition studies (Figure 1F) [49,50]. These materials demonstrate the diverse roles that core/support materials play in tailoring the properties and applications of MIPda-based materials. Hollow MIPda was also reported by forming PDA on silica NPs and was followed by their removal via hydrofluoric acid. However, hollow MIPda exhibits high non-specific adsorption leading to low imprinting factors [51].

The synthesis methods involve self-polymerization or electropolymerization in the presence of the respective templates. The resulting MIPda materials exhibit diverse morphologies such as spherical particles, smooth surfaces, compact layers, or rougher surfaces, depending on the specific configuration and core/support material used. Overall, Table 1 showcases the versatility of MIPda in modifying core/support materials, enabling the development of tailored materials with unique properties for various applications such as environmental monitoring, food control, and clinical diagnosis. The Comments column provides further insights and explanations that complement the data presented in the adjacent columns.
polymers-15-03712-t001_Table 1Table 1Different core/support materials are used to be modified with MIPda.Substrate MaterialsConfigurationTemplateSynthesis MethodMorphologyRole of the Core/Support MaterialsCommentsMagnetic NanoparticlesFe_3_O_4_ NPs [34]ThionineSelf-polymerization if the presence of saturated O_2_ solutionuniform spherical particles and smooth surface of Fe_3_O_4_-MIPdaFe_3_O_4_ NPs: 600 nmFe_3_O_4_-MIPda: 600 nm(thin layer)NanozymeMagnetic propertiesFe_3_O_4_ NPs were prepared by the solvothermal methodFe_3_O_4_@SO_2_ [35]ovalbuminSelf-polymerizationspherical and relativelyuniformMNPs: 13 nmFe_3_O_4_@SiO_2_: 73 nmFe_3_O_4_@SiO_2_@MIPda: NAMagnetic propertiesSiO_2_ provides hydroxyl groups and protects Fe3O4 from oxidation and acid attack during the removal of templatesFe_3_O_4_@fibrous SiO_2_ [36]lysozymeSelf-polymerizationThe MIPDA layer evenly covered the fibrous SiO_2_ surface. The average diameter of the MIP-lysozyme microsphere is 445 nm, indicating an average PDA imprinting layer thickness of 45 nmMagnetic properties and high surface area (570 m^2^/g) allowing for fast adsorptionThe PDA layer exhibits a photothermal effect and shows the controlled release property triggered by NIR laserFe_3_O_4_/PMMA [37]lysozymeSelf-polymerizationFe_3_O_4_: 6 nmFe_3_O_4_/PMMA: 150 nmFe_3_O_4_/PMMA/MIP; 180 nmMagnetic properties and hydroxyl groupsFe_3_O_4_/PMMA was prepared via miniemulsion and the lysozyme was immobilized on the surface of Fe_3_O_4_/PMMA. However, PMMA is not good support for immobilizing lysozymeMetal/metal oxideCdSe–CdS–Zn/Ti substrate [39]L-phenylalanineElectropolymerizationCdS/CdSe layer: spherical structure; CdS/CdSe–MIPda: spherical smooth ad compactPhotoelectrochemical propertiesThe preparation of the electrode is complicatedglassy carbon electrode modified with AuNPs [40]Pseudomonas aeruginosaelectropolymerizationuniform MIPAuNPs were employed as a specialized intermediary and interface to enhance the loading rate of the aptamer sequenceThe dual precise molecular recognition characteristics of both MIP and aptamers resulted in exceptional sensing capabilitiesAuNP-coated screen-printed carbon electrode [41]The igE-binding epitope of ovalbuminelectropolymerizationspherical-like clusterelectrocatalyticactivityThe limited electroactivity of MIPda was countered by electro-depositing AuNPs onto the electrode surface before imprintingcarbon fibers modified with Fe_3_O_4_/MnO_2_ nanoparticles [32]CarcinoembryonicAntigenSelf-polymerizationnanosheetMnO_2_ nanosheets were incorporated into this design due to their ability to enhance the affinity and biocompatibility of the supports through interactions with proteinsManageable dimensions and structure, a significant surface area, as well as straightforward surface modificationAuNPs-CNT-MIPda [43]ureaElectropolymerizationNAConductive propertiesA covalent bond is established between the numerous AuNPs and the thiol groups present on the aptamerGold electrode [42]SulfamethoxazoleElectropolymerizationNAConductive properties--Carboneous materialsmulti-walled carbon nanotubes [44]sunset yellowSelf-polymerizationThe thickness layer of MIP: ~1.8 nm The thickness layer of NIP: ~3.5 nmConductive propertiesThe difference between the thickness layers of MIP and NIP indicates that sunset yellow inhibited the self-polymerizationof dopamine, to some extent, on the MWCNT surfacePencil graphite electrode [46]malathionElectropolymerizationThe roughness of the electrode increased its modification with MIPElectrode supportIn this work, a peptide nanotube (functionalized PDA-based and molecularly imprinted) was employedScreen printed carbon electrode—Reduced graphene oxide [45]Epitope of gliadinElectropolymerization--Reduced graphene oxide enhances the sensitivity of the sensorReduced graphene oxide improved the speed of electron transfer kinetics, contributing to a lowered detection limit and an extended linear detection rangeQuantum dotsQuantum dots [47]rabbit IgGSelf-polymerizationQDs-MIPda have spherical shapes as Quantum dots except with a rougher surfacesuperb optical properties and the ability to multiplexDopamine was polymerized in the presence of Caffeic acid to offer carboxyl groupssilica NPssilica NPs [48]Sunset yellowSelf-polymerizationSiO_2_ NPs::smooth surface and an average diameter of ~15 nm.The MIPda layer thickness is ~5.5 nm The NIPda layer thickness is ~8.0 nmsupporting matrice to prepare surface imprinted PDADopamine adheres to the surface of SiO_2_.The difference between the thickness layer of MIPda and NIPda indicates that sunset inhibits the self-polymerization of dopamine, to some extent, on the SiO_2_ surfaceHollow (Silica NPs were removed) [51]horseradish peroxidaseSelf-polymerizationA very thin layer of MIPdasacrificial matrixThe imprinting factor of the hollow MIP was better than that of the solid MIPsQCM crystalQCM crystal [49]Pepsin, bovine serum albumin, human serum albumin, and lysozymeSelf-polymerizationNASupport ofQuartz crystal microbalanceThe protein recognition on MIPda-functionalized QCM crystals depended on both the match between recognition sites and the target protein, as well as non-specific interactions between proteins and the MIPda filmQCM crystal [50]Hepatitis B antigenSelf-polymerizationThe bumpy appearance of the MIPda-QCM crystalSupport ofQuartz crystal microbalance--PMMA: Polymethyl methacrylate.
Figure 1Various MIPda composites including (**A**) Fe_3_O_4_@MIPda (reprinted from reference [34] with permission of Elsevier), (**B**) SPCE/AuNPs/MIPda (reprinted from reference [41] with permission of Elsevier), (**C**) Gold electrode/MIPda (reprinted from reference [42] with permission of Elsevier), (**D**) MWCNTs/MIPda (reprinted from reference [44] with permission of Elsevier, (**E**) SiO_2_@MIPda (reprinted from reference [48] with permission of Elsevier), (**F**) QCM crystal/MIPda (reprinted from reference [49] with permission of Elsevier).
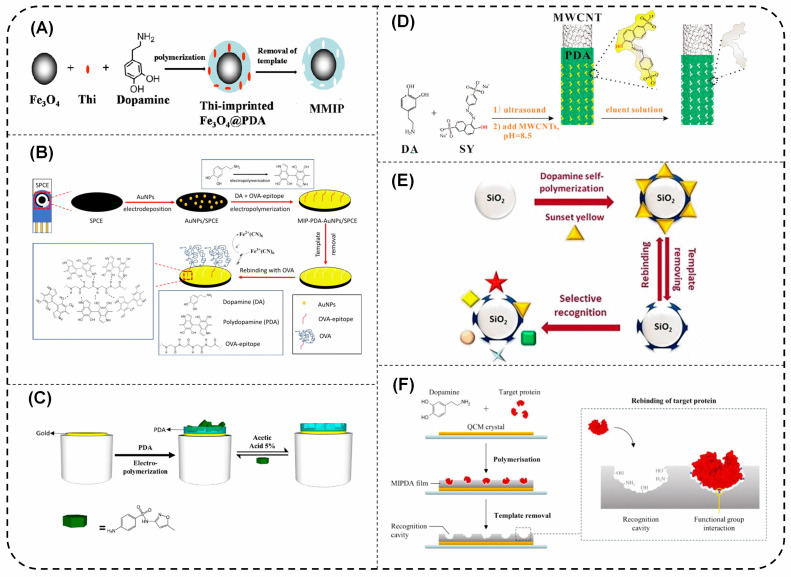



## 3. Types of Templates

Dopamine was applied for various templates such as ions, small molecules, epitopes, proteins, and viruses. Table 2 provides information about the use of MIPda for various templates, including ions, molecules, epitopes, proteins, and viruses. The table includes details about the template family, the specific template used, the removal solution employed, the analytical method applied, and the imprinting factor (IF) obtained for each case.

### 3.1. Ions

Dopamine has gained significant attention as a monomer in the synthesis of ion-imprinted polydopamine (IIPda) due to its unique chemical properties and biological relevance. IIPda is a versatile material that can selectively bind specific ions, making it useful for various applications. For example, IIPda was developed for the electrochemical detection of Nickel ions [52].

The use of dopamine as a monomer in IIPda offers several advantages. Firstly, dopamine possesses both catechol and amine functional groups, which can form strong coordination bonds with metal ions through complexation reactions. This allows for the incorporation of metal ions into the polymer matrix during the synthesis process. Additionally, dopamine is known for its self-polymerization ability, which simplifies the preparation of IIPda without the need for complex synthetic procedures.

The resulting IIPda materials exhibit remarkable properties. The polymer matrix provides structural stability and mechanical robustness, while the imprinted metal ions confer specific recognition sites within the material. This molecular imprinting process leads to the creation of binding cavities that are complementary, in shape, size, and chemical functionality, to the target ions.

However, while the advantages of using dopamine as a monomer in IIPda are evident, it is crucial to acknowledge the broader context. One notable challenge emerges from the relative scarcity of works focusing on IIPs, which may indicate certain complexities in the synthesis process. For instance, the removal of ions from polydopamine matrices often requires an acid medium, which, while effective for ion removal, can also impact the structure of polydopamine. This indicates the need for the further exploration and optimization of synthesis conditions to mitigate potential drawbacks.

### 3.2. Molecules

Dopamine has been explored for the creation of MIPda. Dopamine possesses functional groups such as catechol and amine, which can form various types of interactions, including hydrogen bonding, electrostatic interactions, and π-π stacking, with the target molecule during the polymerization process. This allows for the creation of specific binding sites that are complementary to the shape, size, and chemical functionality of the target molecule.

Dopamine has been exploited to prepare MIP for a large variety of small molecules such as sulfamethoxazole [42], diethylstilbestrol [53], gallic acid [68], sunset yellow [44], bisphenol A [58], glucose [31], cinnamic acid [69], and chlorpyrifos [30]. However, some molecules may inhibit the polymerization of dopamine leading to PDA-based materials with smaller sizes and higher surface area compared to non-imprinted PDA [42,44,48,56]. It should be noted that MIP and NIP must have the same thickness [59].

MIPda was employed to discern the chirality of molecules. In this regard, Arabi et al. proposed a recognition mechanism known as the “inspector” recognition mechanism (IRM) [55]. This mechanism was implemented using a chiral imprinted PDA layer, which was coated on a surface-enhanced Raman scattering (SERS) tag layer. The IRM operates by detecting the permeability change in the imprinted PDA layer following chiral recognition. An inspector molecule is used to scrutinize the permeability, enabling the IRM to prevent the nonspecific recognition of imprinted materials and achieve absolute chiral discrimination. Leibl et al. applied the computational design (density functional theory) using the binding energies for the complex formation of dopamine with different nitro-explosive components such as 3-thiophene acetic acid, triacetone triperoxide, trimesic acid, and 2,4,6-trinitrotoluene [59]. However, dopamine rearranges during polymerization, leading to a lack of stability in the dopamine–template complex.

Utilizing virtual template molecules that share a similar chemical structure to the target molecules can prevent the issues arising from template molecule leakage. Miao et al. demonstrated the usefulness of bisphenol-A as a dummy template for Dichlorodiphenyltrichloroethane [58].

### 3.3. Epitopes

PDA coatings have been utilized as versatile platforms for the immobilization of proteins, offering robust and adhesive surfaces that facilitate the stable attachment and biofunctionalization of various biomolecules [70]. This feature, among others, has granted PDA a wide range of applications in the MIP for biological compounds. Indeed, dopamine serves as an advantageous monomer for epitope-imprinted polydopamine (EIPDA) due to its self-polymerization ability, functional versatility, and specific interactions with target epitopes. For instance, dopamine has been used to construct EIPDAs for specific epitopes such as ovalbumin IgE-binding epitope [41] and gliadin epitope [45]. In one study, the EIPDA showed superior affinity (IF = 8.68) for the gliadin epitope compared to the parent protein, gliadin [45]. This enhanced binding can be explained by the restrictions in diffusion for the original protein during the rebinding process, combined with competition from nonspecific binding sites on the PDA film. Notably, the obtained IF value for the dopamine-based EIPDA was higher than that achieved with a Bis-(bithiophene)-based MIP using the same epitope [71].

### 3.4. Proteins

PDA exhibits numerous advantageous characteristics that make it an exceptional polymer system for protein imprinting. Its structure encompasses a wide array of functional groups, including amino groups, hydroxyl groups, and π–π bonds. These functional groups can be effectively utilized for protein recognition in MIPs. One of the key benefits of PDA is its high hydrophilicity, enabling its synthesis in an aqueous environment that is compatible with the protein templates. This aqueous synthesis provides a favorable setting for incorporating proteins into the polymer matrix, ensuring their proper imprinting and retention of their structural features. Dopamine has been employed to prepare MIPs for a large variety of proteins such as Pepsin [49], carcinoembryonic antigen [32], Thionine [34], rabbit IgG [47], human serum albumin [61], Lysozyme [62], Horse radish peroxidase [51], bovine hemoglobin [63], and Papain [65]. The column of imprinting factor shows that this key parameter usually comprises between 4 and 6, which appeared higher than MIPda for molecules. The low imprinting factor observed with the MIPda for horse radish peroxidase is due to the hollow structure of MIPs [51]. Indeed, the multifunctional groups in the PDA network led to the high binding of horseradish peroxidase on the hollow NIPs. In some works, authors did report the imprinting factor and they evaluated the MIPda through a signal comparison with NIP [47,65]. However, it is intriguing to observe, in some papers, the absence of a signal response in the NIP [34]. In these cases, the NIP layers’ thicknesses are so large that the passage of electrons is impeded. These results are usually observed when the diameter of MIP particles or the thickness of the layer of MIPda is much lower than NIP, leading to high binding capacities of MIPda compared to NIPda, and this difference is not due to the imprinted cavities of MIPda.

In the process of polymerization, dopamine monomers tend to gather around protein templates through interactions involving functional groups, such as hydrogen bonds and π–π interactions. This leads to the incorporation of protein templates within the resulting PDA film. MIPda offers amino groups that adsorb, selectively and non-selectively, protein templates, leading to non-specific adsorption. Han et al. introduced a straightforward approach to mitigate nonspecific adsorption by enveloping the imprinted PDA coatings with mildly crosslinked nonlinear poly(ethylene glycol) (PEG) layers through aqueous precipitation polymerization prior to removing the template [62].

### 3.5. Viruses and Bacteria

The bio-inspired PDA chains displayed non-covalent functionalities, including amine, hydroxyl groups, and π–π bonds, which played a role in selectively binding with the amino acids of proteins on the virus surface. Dopamine possesses redox-active functional groups, such as phenolic hydroxyl and catechol moieties. These groups can facilitate the incorporation of target virus-related molecules during the imprinting process. Moreover, PDA has been extensively studied for its biocompatibility and low cytotoxicity, making it suitable when designing virus-imprinted materials intended for diagnostic or therapeutic purposes. In the self-polymerization under mild conditions of dopamine, forming PDA films or nanoparticles is convenient for imprinting viruses. For instance, the whole hepatitis A virus was imprinted on SiO_2_@PDA [66] and Fe_3_O_4_@PDA [67] while, in another works, the authors reported the imprinting of the hepatitis B core antigen [50]. The MIPda exhibited a higher response than NIP but the removal solution was composed of SDS and acetic acid which may activate the surface of PDA, leading to the unsuccessful creation of imprinted cavities. The bacterium-imprinted PDA has not been reported yet but the results obtained for viruses are an incentive to apply dopamine in the preparation of MIPs for bacteria.

## 4. Preparation Methods of MIPda

### 4.1. Self-Polymerization

The self-polymerization of dopamine is the most used technique for the synthesis of MIPda with a percentage of 65%. Dopamine, a small molecule, can undergo oxidative polymerization in the presence of a basic medium and room oxygen. The use of an oxygen-saturated solution [34] and purging air into the system was also applied [36]. The process begins with the oxidation of dopamine, which generates reactive intermediates, including quinones, leading to the formation of covalent bonds between dopamine molecules. As the polymerization progresses, template molecules are introduced into the reaction mixture, and they interact with the growing PDA network. This interaction results in the creation of specific binding sites within the polymer matrix, which can selectively recognize and bind to the target molecule. After the polymerization is complete, the template molecules are removed from the polymer matrix, leaving imprinted cavities that possess complementary shapes, sizes, and functional groups to the template molecule (Figure 2). The self-polymerization of dopamine offers advantages such as simplicity and versatility. The self-polymerization of PDA can be accelerated using chemical oxidants [72], UV irradiation [72], and microwave irradiation [73]. Ultrasound-assisted synthesis was successfully applied to accelerate the polymerization processes of conventional monomers such as methacrylic acid [16,74]. We have tested the application of this technique to accelerate the synthesis of PDA synthesis, but no acceleration was observed (data not published).

The self-polymerization method can be conducted in batches or by dipping solid support in a solution containing dopamine monomer [55]. Traditional self-polymerization usually requires several hours. The use of initiators to accelerate the self-polymerization of dopamine for the preparation of MIPda was introduced in 2021, achieving MIPda in only 1 h [28].

The self-polymerization of dopamine into PDA occurs using various solvents, such as sodium hydroxide, sodium bicarbonate, PBS, and Tris, resulting in PDA with distinct morphological and physicochemical characteristics associated with each solvent, as demonstrated by Patel et al. when they investigated their effects under similar pH = 8.5 [75]. Nevertheless, when it comes to the self-polymerization of dopamine for MIPda, both tris-buffer [28] and PBS [35] serve as frequently employed mediums. Tris-buffer is particularly well-suited for maintaining a pH of 8.5, while PBS is better suited for a pH of 7.5. These buffer solutions play a crucial role in facilitating the formation of PDA with desirable properties and functionalities.

Over the years, the use of the self-polymerization technique to prepare MIP was not applied only with dopamine monomer but also with other catecholamines such as norepinephrine [76,77]. In a recent development, the self-polymerization technique was successfully extended to include the 5,6-dihydroxy-1H-benzimidazole (DHBI) monomer [78]. Remarkably, DHBI exhibited comparable reaction pathways to dopamine, leading to the formation of a lightly cross-linked, p-conjugated poly(DHBI). Notably, the polymerization of DHBI demonstrated an accelerated rate compared to dopamine, and interestingly, it can be further enhanced under UV stimulation. This expansion of monomer options not only demonstrates the versatility of the self-polymerization approach but also enriches the diversity of properties of the resulting self-polymerized MIPs, opening new possibilities for adaptation to a wide range of templates.

### 4.2. Electropolymerization

The electrochemical synthesis of MIPda is a technique that utilizes an electrochemical process to fabricate MIPs. The process involves the electrooxidation of dopamine monomers on an electrode surface to initiate the polymerization and imprinting of target molecules. Typically, a conductive substrate, such as a glassy carbon electrode, is used as the working electrode. Dopamine monomers are dissolved in a suitable electrolyte solution, and the electrode is immersed in this solution. When an electrical potential is applied to the electrode, dopamine undergoes electrooxidation, leading to the formation of a PDA film on the electrode surface. During this electrochemical polymerization, template molecules are introduced into the system. The dopamine monomers polymerize around the template molecules, resulting in the formation of imprinted sites that possess a specific affinity for the target molecule (Figure 2).

The electrochemical synthesis of MIPda offers advantages such as precise control over the film thickness, improved accessibility of the imprinted sites, and the ability to integrate the MIPDA films onto various electrode surfaces for sensor and sensing applications.

Table 3 provides a summary of the various electrochemical methods used for the synthesis of imprinted PDA and their corresponding details. The table presents different preparation methods, core materials, templates, electrochemical conditions, sensors used, and additional comments for each method. The Comments column serves as a valuable addition to help provide context and insights that complement the data presented in the other columns. The first entry describes the use of an Au electrode and immunoglobulin G as the template, with specific electrochemical conditions applied in a PBS buffer solution [79].

Before the synthesis of MIPda, the immobilization of IgG on the electrode surface takes place. Subsequent entries highlight diverse approaches, including the use of sulfamethoxazole [42], an amino-aptamer for 2,4,6-trinitrotoluene [80], uric acid [60], urea–aptamer complex [43], L-phenylalanine [39], carboxylic-acid-based structural analogs [59], and the ovalbumin IgE-binding epitope [41] as templates. Various core materials, such as Au nanoparticles, GCE modified with AuNP@fullerene, and nickel nanoparticles wrapped with carbon, are utilized in the electrochemical synthesis. The sensors employed for detection purposes range from QCM sensors to amperometric and impedance spectroscopy-based techniques. The comments provided in Table 3 shed light on aspects such as imprinting efficiency, the ease of the preparation method, and the optimization of the electrochemical parameters.

The number of electropolymerization cycles has a significant effect on the imprinting process. The development and sensing efficacy of the imprinted polymeric membrane hinged on the number of scan cycles [81]. A low cycle count resulted in a delicate and easily damaged MIP film during template removal. Moreover, an insufficient number of recognition sites formed [82]. Conversely, an excessive electropolymerization yielded a thick PDA layer that hindered template removal. Additionally, excessively coating the non-conductive PDA compromised electron transfer and sensitivity [83].

On the other hand, the pH of the electropolymerization solution plays an important role in MIPda-based electrochemical sensors. The pH of the MIP solution has a significant impact on the electrode sensing performance. At high pH levels, the phenolic group in PDA becomes deprotonated while, at low pH levels, the amino group becomes protonated. This property can influence PDA synthesis, necessitating pH optimization of the MIP solution. The findings revealed that the highest current values were observed at pH 7.5 and 8.0. While Yang et al. achieved the thickest PDA growth at pH 8.5, the resulting MIP PDA lacked satisfactory sensitivity [65]. Recent studies indicated that a pH of 8.5 or higher could trigger premature, uncontrolled DA polymerization before surface synthesis [65]. Table 3 provides information about the electrochemical methods used for the synthesis of MIPda, including the core materials used, templates, electrochemical conditions, sensors employed, and additional comments.

Self-polymerization may interfere with electropolymerization, particularly when electropolymerization is conducted in the basic medium at a low rate. However, the researchers avoid this interference by bubbling N_2_ to remove oxygen from the electropolymerization solutions [46,54].

As shown in Figure 2, the self-polymerization (65%) of dopamine offers versatility in preparing MIPda composites, including core-shell structures and nanocomposites. This allows integration into various type of sensors including optical and electrochemical sensors and SPE methods. In contrast, electropolymerization (35%) mainly focuses on electrochemical sensors by depositing PDA films on electrodes. While it provides control over film thickness and site accessibility, it is mainly used for electrode preparation.

## 5. Characteristics of MIPda

### 5.1. Structure of PDA

The structure of PDA remains highly intricate and, as of yet, a definitive and comprehensive understanding of its formation and complete structure has not been conclusively established. Despite numerous proposed structures and mechanistic pathways, as presented in the literature [84], a clear mechanism for its formation still eludes researchers. Researchers have put forth various theories about the composition and structure of PDA. Some studies suggest that PDA is composed of covalently linked dihydroxy indole, indoledione, and dopamine units [85]. However, an alternative structural model has been proposed recently, suggesting that the dihydroxyindoline, indoline-dione, and dopamine units are not covalently linked but instead held together through hydrogen bonding between oxygen atoms or π stacking interactions [84]. Furthermore, Liebschera et al. developed another structural model, proposing mixtures of different oligomers in PDA, where indole units with varying degrees of (un)saturation and open-chain dopamine units coexist [86].

### 5.2. Morphology

Surface roughness plays a critical role in characterizing the properties of MIPda. To investigate surface morphologies, atomic force microscopy (AFM) was utilized to analyze MIPda-coated and NIP-coated quartz crystal microbalance (QCM) crystals [50] and SiO_2_ NPs [48]. The root means squared surface roughness (Rq) measurements were employed to monitor morphological changes on the surface of materials. Notably, the MIPda-QCM and MIPda-SiO_2_ exhibited a higher Rq value, suggesting a rougher surface compared to the NIP-QCM crystal and NIP-SiO_2_. Moreover, the surface of MIPda-SiO_2_ and NIP-SiO_2_ are rougher than the SiO_2_ surface, confirming the well decoration of PDA on the SiO_2_ [48]. These findings emphasize the influence of surface functionalization with the MIPda film on the overall roughness, highlighting the importance of surface roughness in characterizing MIPDA-based materials. The MIPda adopts various core/support shapes, including spherical(Figure 3a,b) [34], film (Figure 3c,d) [49], nanosheet (Figure 3e,f) [32], and nanotube (Figure 3g) [44] shapes.

The Fe_3_O_4_-MIPda image (Figure 3a) appeared smoother in comparison to the image of Fe_3_O_4_ NPs (Figure 3b). The QCM crystals (Figure 3c) with the coating exhibited a slight blurriness when contrasted with the bare QCM crystal (Figure 3d). This effect is likely attributed to charging artifacts arising from the deposition of non-conductive MIPda films on the surface. Upon applying the MIPda layer coating, there was a notable alteration in the surface morphology, transforming the previously gully-like surface into a granular texture (Figure 3e,f).

MIPda’s properties are intricately linked to the thickness of its layers. Typically, MIPda exhibits an ultra-thin layer, with thicknesses measuring less than 10 nm [44,48,51], making it highly promising for precise drug delivery and molecular recognition applications. Surprisingly, researchers have also observed instances of relatively thicker MIPda layers [36,37], which further expands its potential for versatile use. The adaptability of MIPda in different shapes and layer thicknesses opens up exciting opportunities for advancing drug delivery systems and molecular sensing technologies.

### 5.3. Wettability

During the polymerization process of dopamine to form PDA, various functional groups, such as amine and hydroxyl, are incorporated into the material’s structure, leading to its hydrophilic nature. These functional groups facilitate hydrogen bonding and other interactions with water molecules. These interactions create a conducive environment for the diffusion of target analytes through the MIPda material, enhancing the accessibility of the imprinted cavities to the target molecules. This improved accessibility is essential for the efficient binding and recognition of the target analytes. Water contact angle measurements serve as a valuable tool for characterizing the wettability of MIPda. These measurements play a crucial role in validating the successful modification of substrate materials with MIPda films. Following surface functionalization, the contact angles of MIPda-based materials exhibit a notable decrease compared to bare materials, highlighting the enhanced wettability conferred by MIPda. This decrease in contact angles indicates an increase in the hydrophilicity of the substrate surface. The improved hydrophilicity can be attributed to the presence of hydroxyl groups within the PDA structure [50], along with the contribution of hydrophilic quinone/hydroquinone moieties present in the MIPda material [87]. These combined factors work synergistically to create a surface that strongly interacts with water molecules, holding significant implications for various applications where controlled wettability and specific molecular interactions are essential.

## 6. Applications of MIP-PDA in Sensors

MIPda has emerged as a groundbreaking material that significantly improves molecular-recognition-based sensing approaches. MIPda is a synthetic polymer with exceptional affinity and selectivity for specific target molecules, making it an ideal candidate for sensor applications. The unique properties of MIPda, such as its robustness, biocompatibility, and ease of synthesis, have fueled its integration into various sensing platforms, including optical, electrochemical, and SPE coupled to sensors. This introductory paragraph delves into the diverse applications of MIPda in these sensor technologies, showcasing its remarkable versatility and potential to revolutionize a wide array of fields, ranging from healthcare diagnostics and environmental monitoring to food safety and beyond. Through a comprehensive exploration of MIPda’s multifaceted applications, this article seeks to shed light on its immense impact and promise in advancing the landscape of molecular-recognition-based sensing systems.

### 6.1. MIPda-Based Optical Sensors

MIPda-based optical sensors have found diverse and impactful applications thanks to their unique combination of molecular imprinting and optical sensing. For instance, a microfiber interference sensor based on MIPda was developed by Liu et al. for the specific detection of C-reactive protein [88]. Unlike traditional imprinting methods, this approach induced the rearrangement of the template molecule during dopamine self-polymerization, creating complementary hydrophobic/hydrophilic and charge distributions in addition to shape and size within the imprinting cavity. By combining this approach with optical fiber interferometry, the optical sensor demonstrated a significantly low LOD of 5.8 × 10^−10^ ng/mL, outperforming commercial ELISA kits by eight orders of magnitude. The CMIP-PDA microfiber sensor exhibited strong repeatability, high selectivity, and the potential for ultrasensitive, label-free CRP diagnosis. This innovative technique opens doors to quantitatively monitoring biomarkers with extremely low concentrations and has broader applications for detecting various biomolecules.

Lu et al. developed a unique approach using MIPda in a sandwich structure, combined with dual signal amplification using MnO_2_ nanosheets and MoS_2_ nanoflowers, to construct a fluorescence sensor for the ultrasensitive detection of trace amounts of carcinoembryonic antigen (CEA) [32]. The sensors utilize a combination of MIPda, MnO_2_ nanosheets, and MoS_2_ nanoflowers to achieve high sensitivity and specificity. The sensors can detect CEA concentrations in a linear range of 0.01 ng/mL to 10 μg/mL, with a remarkably low LOD of 3.5 pg/mL. This work holds promise for improving the early detection of CEA-associated cancers.

Another interesting approach was introduced by Wang et al. for the detection of antibiotics using an optical sensor based on SPR [89]. This study aimed to create a localized surface plasmon resonance (LSPR) biosensor for swift, sensitive, and specific enrofloxacin (ENRO) detection, using MIPda as the recognition element. PDA-MIP film was formed on an LSPR sensor chip’s surface through dopamine and ENRO polymerization. The sensor selectively captured ENRO post-blocking and removal steps. To enhance detection signals, protein-conjugate competitors amplified LSPR signals. Detection took only 20 min, with a range of 25–1000 ng/mL. MIPda film showed higher ENRO binding than its non-imprinted counterparts, distinguishing ENRO from analogs. With high sensitivity, specificity, reusability, and stability, the developed LSPR/MIPda sensor holds the potential for rapid in-field ENRO residue detection.

Table 4 provides an overview of the key research works on MIPda-based optical sensors. It highlights target analytes, detection methods, and significant features of each study. MIPda-based optical sensors show great promise in enhancing detection precision and sensitivity. These sensors use tailored molecular imprinting to bind specifically with target molecules, allowing accurate measurement. PDA integration adds versatility, forming recognition sites through polymerization. Successful in various fields, like environmental monitoring and medical diagnostics, these sensors utilize light–MIP interactions for easily measurable optical changes. Real-time, label-free detection makes them invaluable for rapid and precise analyses. Continual technological progress ensures their potential for diverse applications in complex analytical scenarios.

### 6.2. MIPda-Based Electrochemical Sensors

MIPda is emerging as a promising functional material for designing highly selective electrochemical sensors [41,44,54,56,59,90,91]. The unique properties of MIPda, including facile preparation, good stability, biocompatibility, and abundant functional groups, make them suitable for molecular imprinting/recognition and electrochemical sensing applications. MIPda can be electrodeposited on conductive surfaces modified electrodes by the one-step self-polymerization of dopamine in the presence of a template molecule. This creates selective recognition cavities in the polymer matrix complementary to the template in shape, size, and functional groups. Recently, MIPda has found many applications in the field of electrochemical sensing. Indeed, some of the key applications of the MIPda include the use of these sensitive materials for the electrochemical sensing of nitro-explosives, the detection of small molecule food contaminants, and the diagnosis of infectious diseases. In this context, Li et al. have created an electrochemical sensor to detect illicit stimulants MDA and MDMA (ecstasy) [54]. They used a MIPda film on a gold electrode, prepared via electrochemical polymerization. The film had specific binding sites for MDA and MDMA, leading to strong detection. The sensor showed excellent performances with low detection limits (37 nM for MDA, 54 nM for MDMA) using differential pulse voltammetry. It proved selective, stable, reproducible, and effective with real urine samples. This sensor could be a rapid diagnostic tool for detecting MDA and MDMA abuse in drug investigations.

Another interesting application of MIPda in electrochemical sensing has been introduced by Yin et al. for sunset yellow sensing [48]. They utilized self-polymerized PDA in water to form an MIP on multi-walled carbon nanotubes (MWCNTs), using sunset yellow as a template. The resulting nanocomposites were examined for their electrochemical response to sunset yellow (Figure 4). The modified electrode displayed precise and sensitive detection of sunset yellow due to well-matched cavities on MWCNTs and the blocking effect of non-imprinted PDA. The sensor showed a linear relationship from 2.2 nM to 4.64 μM, with a low LOD of 1.4 nM. It proved to have selectivity, stability, reproducibility, and successful real-sample detection. This technique holds promise for other PDA-based MIP sensors and practical uses.

Recently, the detection of nitro-explosives has been explored using the MIPda-based electrochemical sensor. In this context, Leibl et al. developed a sensitive electrochemical sensor to detect nitro-explosives in water using thin MIPda films [59]. The MIP films were electropolymerized on gold electrodes through cyclic voltammetry, with dummy templates to mimic the nitro-explosive molecules TNT and 1,3,5-trinitroperhydro-1,3,5-triazine. These imprinted films enhanced sensitivity 105-fold compared to unmodified gold electrodes. The MIP films concentrated the target molecules near the transduction element, resulting in improved sensitivity. The MIP films showed reproducible binding in a phosphate buffer, with a dynamic detection range of 0.1 nM to 10 nM for both TNT and 1,3,5-trinitroperhydro-1,3,5-triazine. They also exhibited increased selectivity over similar related compounds.

MIPda has found application in protein detection, especially for Ovalbumin protein. Indeed, in a recent study reported by Khumsap et al., a precise electrochemical sensor was developed by imprinting PDA with the OVA IgE-binding epitope to detect ovalbumin [41]. They optimized various factors in the process and used differential pulse voltammetry with K_3_Fe(CN)_6_ and KCl electrolyte to detect OVA. The sensor demonstrated remarkable sensitivity with a detection limit of 10.76 nM, a linear range from 23.25 to 232.50 nM, and strong selectivity against other proteins. Successful detection in wine samples showed potential for broader use in identifying allergenic proteins in the food chain (see Figure 1B).

Table 5 summarizes the notable research endeavors involving MIPda-based electrochemical sensors. The table outlines the target analytes, electrode modification methods, detection techniques, and key characteristics of each investigation. Its easy preparation, tunable selectivity, and sensitivity make MIPda a promising interface for the electroanalysis of various chemicals and biomolecules. The imprinting approach can be further explored to improve the analytical performance of PDA-based electrochemical sensors. These applications demonstrate the versatility and potential of MIPda in electrochemical sensors for various analytical purposes.

### 6.3. Solid-Phase Extraction Coupled to Sensors

MIPda combined with SPE forms a synergistic approach that enhances the selectivity and efficiency of analyte extraction and detection [61,69]. MIPda provides specific binding sites tailored for the target molecule, improving the extraction process’ selectivity. When integrated with SPE, the MIPda acts as a specialized sorbent, selectively capturing the target analyte from a complex sample matrix. This combination enhances pre-concentration, reduces interference, and improves overall sensitivity in analytical methods. A result is a powerful tool for the extraction and detection of specific compounds in diverse fields, such as environmental monitoring, pharmaceutical analysis, and food safety, where accuracy and precision are essential [69].

Yin et al. presented a new approach involving magnetic imprinted polymers coated with cinnamic acid polydopamine, designed for the simultaneous and selective extraction of cinnamic acid, ferulic acid, and caffeic acid from radix scrophulariae samples [69]. These innovative magnetic imprinted polymers were created through surface imprinting polymerization, utilizing magnetic MWCNTs as support material, cinnamic acid as the template, and dopamine as the functional building block. The analysis demonstrated excellent magnetic properties, a high adsorption capacity, selectivity, and rapid binding kinetics for cinnamic acid, ferulic acid, and caffeic acid. By coupling with high-performance liquid chromatography, we extensively investigated the magnetic imprinted polymers’ use as a magnetic SPE sorbent under varying extraction conditions. Successfully employed for the purification and enrichment of cinnamic acid, ferulic acid, and caffeic acid from radix scrophulariae extracts, the proposed imprinted magnetic SPE method achieved recoveries of 92.4–115.0% for cinnamic acid, 89.4–103.0% for ferulic acid, and 86.6–96.0% for caffeic acid.

Another interesting application of MIPda in SPE, as reported by Yin et al., is for protein capture [61]. Indeed, an innovative and cost-effective approach was developed to imprint proteins onto magnetic MMWNTs surfaces. Human serum albumin (HSA) was used as the template, while dopamine served as the functional monomer. The maximum adsorption capacity for HSA by the magnetic imprinted polymers was determined to be 66.23 mg g^−1^, and equilibrium adsorption was achieved within 20 min. The MIPda demonstrated remarkable selective adsorption capabilities, specifically for HSA. When combined with high-performance liquid chromatography (HPLC) analysis, the magnetic imprinted polymers were effectively employed for SPE and the detection of HSA in urine samples, yielding successful recoveries ranging from 91.95% to 97.8%.

Recently, Elfadil et al. have introduced a study for extracting and quantifying erythrosine B (ERT-B) in food samples [27]. The authors synthesized a composite material comprising MIPda coated onto magnetic nanoparticles (Fe_3_O_4_@PDA@MIP) through an environmentally friendly process (Figure 5). This composite facilitated the magnetic dispersive solid-phase extraction (MDSPE) of ERT-B, ensuring rapid extraction, reusability, and effective pre-concentration. Notably, Fe_3_O_4_@PDA@MIP exhibited a strong imprinting factor (3.0 ± 0.05), showcasing remarkable selectivity against interfering substances such as patent blue and other food matrix components. The MDSPE approach was paired with a smartphone-based colorimetric detection method, yielding performance akin to UV–visible spectroscopy detection. The achieved reliable ERT-B quantification within the range of 0.5–10 mg/L, with a detection limit of 0.04 mg/L. The developed method was successfully applied to determine ERT-B content in various food samples, including juice, candy, and candied cherries, yielding satisfactory recovery values (82–97%).

Table 6 offers a concise compilation of the research efforts related to MIPda-based SPE. It details the target analytes, extraction methods, detection strategies, and noteworthy aspects of each study. The integration of MIPda into SPE has demonstrated remarkable potential and versatility in enhancing the selectivity, efficiency, and sensitivity of analyte extraction from complex matrices. The tailored molecular recognition properties of MIPda materials provide a unique advantage, allowing for the specific binding and efficient retention of target molecules. This synergy between MIPda and SPE addresses challenges in various fields, ranging from environmental monitoring to pharmaceutical analysis. The resultant MIPda-based SPE methods offer improved accuracy, reduced interference, and enhanced preconcentration capabilities, making them invaluable tools for sample preparation and analysis.

## 7. Challenges and Limitations

Firstly, in the case of general MIPs, the template molecule typically undergoes a self-assembly process where hydrogen bonds form between the specific functional groups of the monomer and the template. However, dopamine oxidizes and rearranges before polymerization, disrupting the interactions between the monomer and the template. This oxidative process hinders the formation of the desired imprinted cavities. Secondly, dopamine exhibits high reactivity, leading to covalent reactions with certain templates during its polymerization under low-alkaline conditions. As a result, the extraction of the template becomes challenging. The strong covalent bonds formed between dopamine and the template impede the efficient removal of the template molecule [28].

Furthermore, while MIPda has demonstrated robustness, biocompatibility, and ease of synthesis, it is essential to recognize that these materials may encounter challenges related to long-term stability. Indeed, PDA is composed of oligomeric species that typically reach only up to the tetramer level [92]. This continuous detachment of PDA compromises the stability and integrity of the imprinted cavities, making them prone to distortion or even destruction.

While dopamine’s unique chemical properties facilitate its self-polymerization, it is important to acknowledge that this may not inherently lead to the same degree of three-dimensionality as observed with conventional cross-linker-mediated imprinting. As such, the resulting molecular cavities in MIPda may exhibit distinct characteristics due to the polymerization mechanism and thin-layer nature of PDA. This aspect underscores the need for a thorough investigation into the dimensionality and stability of the imprinted cavities, which we recognize as an important consideration in the overall framework of our study.

Non-specific binding is a common challenge of MIPs including MIPda. The particular challenge with MIPda is that PDA has important functional groups that can bind several compounds in the matrix rather than the target analyte. PDA can undergo degradation over time, especially under harsh environmental conditions. This can affect the long-term stability and performance of MIPda-based materials in some applications.

While dopamine-based polymers have been successfully employed in various applications, their use may be limited to specific target molecules or classes of compounds. This lack of versatility can restrict the broader applicability of MIPda in diverse fields. In general, the electropolymerization method, including the electropolymerization of dopamine, encounters a challenge when applied to electroactive templates with oxidation potentials falling within the electropolymerization range. This difficulty arises due to the possible oxidation of templates during the electropolymerization process. Furthermore, the self-polymerization of dopamine in the basic medium in the presence of some classes of molecules, particularly phenols, leads to the oxidation of the template molecules.

Lastly, the washing step required to remove the template molecule often involves harsh conditions, including the use of acidic mediums. However, unlike other polymers, PDA is not stable under these acidic conditions [93]. Consequently, the cavities formed within the MIPda can be distorted or destroyed, further undermining the effectiveness of the molecular recognition process. In some cases, the components of the removal solution can interact with the surface of the polymer, leading to its modification and causing high non-specific adsorption, which is another challenge for the removal of templates from MIPs.

To overcome these limitations, ongoing extensive research is exploring alternative monomers and optimization strategies to improve the creation and stability of imprinted cavities using dopamine as a functional monomer.

## 8. Perspectives

Despite the challenges and limitations associated with the use of dopamine as a functional monomer in MIPda, there are several perspectives worth considering for future research and development. Enhancing the stability of MIPda materials is important for long-term performance and practical applications. Researchers can explore strategies, such as cross-linking agents or surface modifications, to improve the stability of MIPda and mitigate the issues related to degradation and detachment. Understanding the specific interactions between dopamine-based polymers and different templates is crucial for achieving high selectivity and avoiding non-specific binding. Further investigations can focus on identifying templates that are compatible with dopamine and on designing tailored monomers to enhance the molecular recognition properties of MIPda. Developing efficient and gentle extraction methods for removing templates from MIPda is essential. Exploring alternative approaches to template removal, such as environmentally friendly solvents or non-destructive techniques, can help preserve the integrity of the imprinted cavities and improve the overall performance of MIPda. Expanding the versatility of dopamine-based polymers and MIPda is another perspective to consider. By exploring different target molecules or classes of compounds and optimizing the synthesis parameters, it may be possible to broaden the scope of applications for MIPda beyond specific targets. This would increase the potential impact of MIPda in various research fields.

In conclusion, despite the challenges and limitations associated with dopamine as a monomer in MIPda, there are promising perspectives for future research. By focusing on monomer optimization, template compatibility, stability enhancement, broadening applicability, and extraction methods, researchers can overcome these limitations and unlock the full potential of MIPda for molecular recognition and other applications.

## Figures and Tables

**Figure 2 polymers-15-03712-f002:**
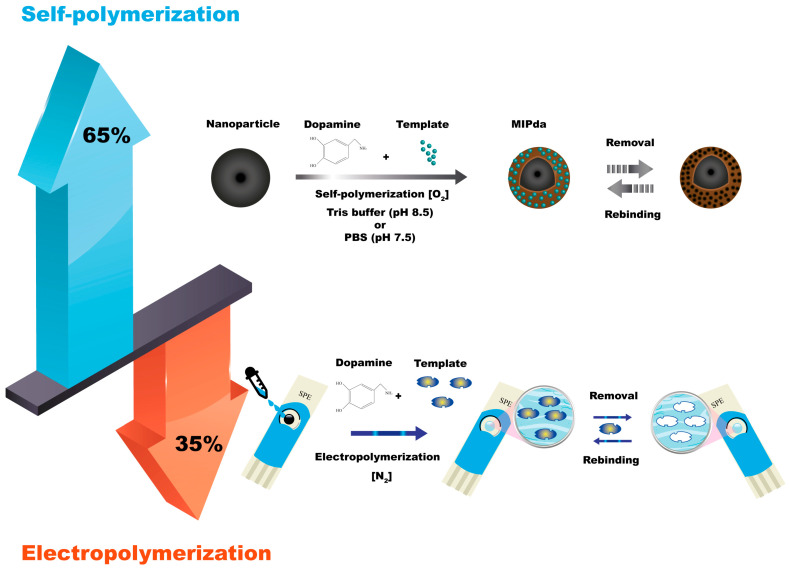
Self-polymerization and electropolymerization methods for synthesis of MIPda.

**Figure 3 polymers-15-03712-f003:**
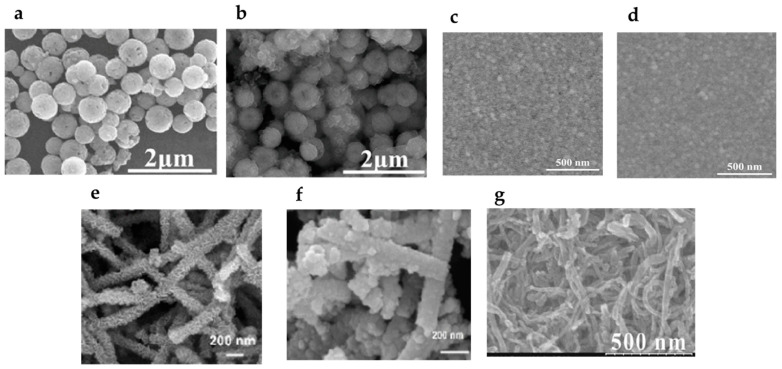
SEM images of (**a**) Fe_3_O_4_ NPs (reprinted from reference [34] with permission of Elsevier), (**b**) Fe_3_O_4_ NPs-MIPda (reprinted from reference [34] with permission of Elsevier), (**c**) QCM crystal (reprinted from reference [49] with permission of Elsevier), (**d**) QCM crystal-MIPda (reprinted from reference [49] with permission of Elsevier), (**e**) carbon fibers/Fe_3_O_4_/MnO_2_ NPs (reprinted from reference [32] with permission of Elsevier), (**f**) carbon fibers/Fe_3_O_4_/MnO_2_ NPs/MIPda (reprinted from reference [32] with permission of ACS), (**g**) MWCNTs-MIPda (reprinted from reference [44] with permission of Elsevier).

**Figure 4 polymers-15-03712-f004:**
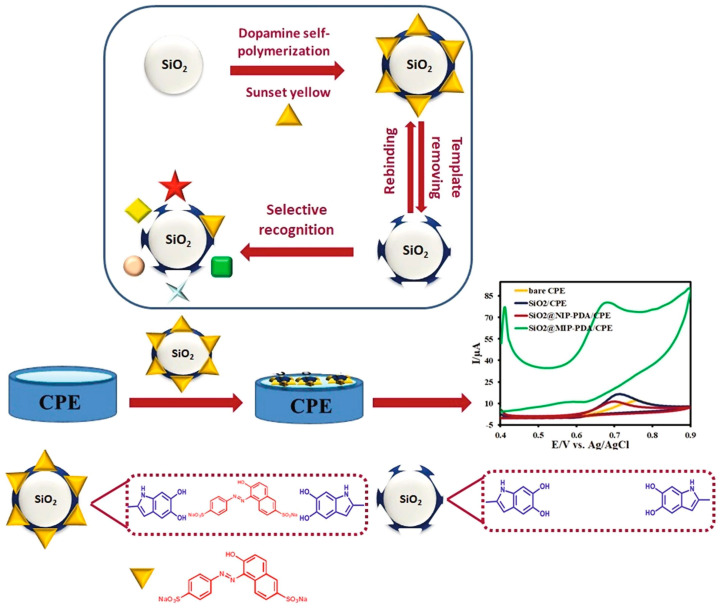
Fabrication process of sunset-yellow-imprinted SiO_2_@PDA NPs electrochemical sensor and its application to detect sunset yellow. Reused, with permission from Elsevier publishers, from reference [48].

**Figure 5 polymers-15-03712-f005:**
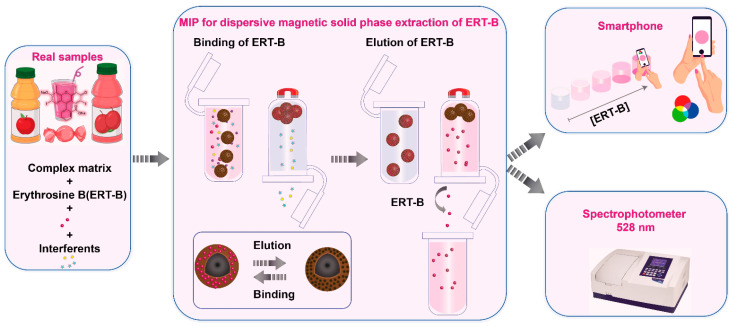
Schematic illustration of the application of magnetic MIPda as an adsorbent in SPE for optical detection of ERT. Permission granted from the corresponding author of reference [27] (MDPI).

**Table 2 polymers-15-03712-t002:** MIPda for ions, molecules, epitopes, proteins, and viruses.

Template Family	Template	Removal Solution	Analytical Method	Imprinting Factor
Ions	Ni^2+^ [52]	0.1 M EDTA (pH = 4.5)	Electrochemical method	N/A
Molecules	Sulfamethoxazole [42]	5% of acetic acid	Electrochemical method	4.84
	Diethylstilbestrol [53]		Electrochemical method	
	3,4-Methylenedioxyamphetamine (MDA) and 3,4-methylenedioxymethamphetamine (MDMA) [54]	0.1 mol L^−1^ NaOH	Electrochemical method	>2
	Tryptophan [55]	0.5% (*v*/*v*) solution of acetic acid	surface-enhanced Raman scattering	N/A
	Tartrazine [56]	*V*_ethanol_: *V*_concentrated ammonia_:*V*_water_ = 7:2:1	Electrochemical method	No response by NIPda and high signal of MIPda
	BPA [57]	10% acetic acid and 20% acetonitrile	Electrochemical method	2.35
	BPA [58]	20% acetonitrile and 3% acetic acid	Electrochemical method	2.6
	Sunset yellow [48]	Ethanol, concentrated ammonia, and water with volume ratios of 7:2:1	Electrochemical method	2.6
	2,4,6-trinitrotoluene and 1,3,5-trinitroperhydro-1,3,5-triazine [59]	Methanol: aqueous ammonia solution at pH 9 (1:1, *v*/*v*)	Electrochemical method	N/A (the NIP-sensor is completely silent and high signal of MIP-sensor)
	L-phenylalanine [39]	N/A	Electrochemical method	17.2
	Uric acid [60]	Methanol/acetic acid (9:1, *v*/*v*)	Electrochemical method	N/A
Epitopes	Ovalbumin IgE-binding epitope [41]	1 M NaOH	Electrochemical method	No response by NIP
	Gliadin epitope [45]	Acetic acid 3% (*v*/*v*) and SDS 1% (*w*/*v*)	Electrochemical method	IF of the epitope = 8.68 IF of the target = 3.14
Proteins	Pepsin [49]	SDS (10%, *w*/*v*) and acetic acid (5%, *v*/*v*)	Quartz crystal microbalance	5.78
	Carcinoembryonic antigen [32]	0.1 M HAc containing 5% SDS (*w*/*v*)	Fluorescence	5.5
	Thionine [34]	acetic acid/acetonitrile solution	Electrochemical method	No response by NIP
	Rabbit IgG [47]	0.1 M glycine-HCL buffer (pH 3.5)	Fluorescence	N/A
	Human serum albumin [61]	0.5 M NaCl	N/A	4.64
	Lysozyme [62]	1% SDS/3% HAc	N/A	6.4
	Lysozyme [37]	SDS and acetic acid (0.1%, *w*/*v*: 3%, *v*/*v*)	N/A	4.38
	Horseradish peroxidase [51]	20% acetic acid	N/A	2.95 (it is not prominent
	Bovine hemoglobin [63]	0.5% triton x-100	Fluorescence	4.1
	Bovine hemoglobin [64]	SDS (10%, *w*/*v*) and acetic acid (10%, *v*/*v*)	N/A	5.33
	Papain [65]	Acetic acid (5%, *v*/*v*) and SDS (10%, *w*/*v*)	Resonance light scattering sensor	N/A (MIP exhibited higher light scattering than NIP)
Viruses	Hepatitis B core antigen [50]	Acetic acid (5%, *v*/*v*) and SDS (10%, *w*/*v*)	Quartz crystal microbalance	N/A (MIP exhibited higher response than NIP)
	Hepatitis A [66]	Acetic acid (5%, *v*/*v*) and SDS (10%, *w*/*v*)	Fluorescence	N/A (MIP exhibited higher response than NIP)
	Hepatitis A virus [67]	Acetic acid (5%, *v*/*v*) and SDS (10%, *w*/*v*)	Fluorescence	N/A (MIP exhibited higher response than NIP)

**Table 3 polymers-15-03712-t003:** Electrochemical methods for the synthesis of imprinted PDA.

Core Materials	Templates	Electrochemical Conditions	Sensors	Comments
Au electrode	Immunoglobulin G [79]	−0.45 and +0.55 Vat a scan rate of 50 mV s^−1^ in PBS buffer Solution	QCM sensor	The igg was immobilized on the electrode surface before the synthesis of MIPda
Au electrode	Sulfamethoxazole [42]	Between−0.6 and 0.6 V at a rate of 20 mv s^−1^ for 60 cycle	Amperometric detection	No significant imprinting and no considerable difference between MIP and NIP
GCE modified with aunps@fullerene	An amino-aptamer for 2,4,6-trinitrotoluene [80]	−0.5 to 0.5 V At 20 mV s^−1^ (13 cycles)	Impedimetric detection	The preparation method is not easy
Nickel nanoparticles Wrapped with carbon	Uric acid [60]	−0.6 to 0.6 V for 10 cycles, scan rate 50 mv/s, and the electrolyte was 0.01 M Phosphate buffer solution (pH 7.4)	DPV	The uric acid can be oxidized during electro-polymerization in the potential range of −0.6 to 0.6 V
AuNPs/CNTs/GCE	Urea–aptamer complex [43]	−0.5 to 0.5 V vs. Ag/Agcl at a scan rate of 20 mV s^−1^ (13 cycles)	Impedance Spectroscopy	The urea was immobilized on the surface SH-AuNPs/CNTs/GCE
Cds/Zn/Ti substrate	L-phenylalanine [39]	+1.5 V to −1.5 V at 50 mV/s, 20 cycles	Photoelectrochemical	The template can be oxidized during polymerization
Au electrode	Carboxylic-acid-based Structural analogs (’dummy’ templates) for nitro-explosives (2,4,6-trinitrotoluene, TNT) And (1,3,5-trinitroperhydro-1,3,5-triazine [59].	−0.5 V and +0.5 V 0.02 V s^−1^ for 15 cycles	--	Dopamine was identified in silico, based on DFT (density functional theory) calculations
Aunp-coated screen-printed carbon electrode	Ovalbumin Ige-binding epitope [41]	−0.5 to +1.0 V at a scan rate of 50 mV/s for 10 cycles	DPV	The pH of solution, concentration of template, and number of cycles of electropolymerization were optimized

**Table 4 polymers-15-03712-t004:** Overview of research works on MIPda-based optical sensors.

Target Analyte	Detection Method	LOD	Notable Features
C-reactive protein (CRP) [88]	Microfiber interference sensor	5.8 × 10^−10^ ng/mL	Low LOD, label-free diagnosis of CRP
Carcinoembryonic antigen (CEA) [32]	Fluorescence sensor	3.5 pg/mL	Sandwich structure, dual signal amplification
Enrofloxacin (ENRO) [89]	Localized SPR biosensor	25–1000 ng/mL	LSPR-based sensor for ENRO residue detection

**Table 5 polymers-15-03712-t005:** Overview of research works on MIPda-based electrochemical sensors.

Target Analyte	Electrode Modification Method	Detection Method	LOD	Notable Features
MDA and MDMA [54]	Electrochemical polymerization	Differential Pulse Voltammetry	37 nM for MDA, 54 nM for MDMA	Rapid detection of illicit stimulants
Sunset Yellow [48]	Self-polymerized PDA on MWCNTs	Electrochemical response	1.4 nM	Precise SY detection, MWCNT cavities
Nitro-explosives [59]	Electropolymerized MIP films	Cyclic Voltammetry	0.1 nM–10 nM	Enhanced sensitivity
Ovalbumin protein [41]	Imprinted PDA electrode	Differential Pulse Voltammetry	10.76 nM	Sensitivity for allergenic protein detection

**Table 6 polymers-15-03712-t006:** Overview of research works on MIPda-based SPE.

Target Analyte	Extraction Method	Detection Method	LOD	Notable Features
Cinnamic acid, ferulic acid, caffeic acid [69]	Magnetic imprinted polymers	HPLC analysis	--	Selective extraction from complex matrix
Human serum albumin (HSA) [61]	Magnetic imprinted polymers	HPLC analysis	--	Protein capture from urine samples
Erythrosine B (ERT-B) [27]	Magnetic dispersive SPE	Smartphone-based colorimetric detection	0.04 mg/L	Selective extraction and colorimetric detection

## Data Availability

Not applicable.

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
