# Peer review of "Unlocking the Potential of Molecularly Imprinted Polydopamine in Sensing Applications"

_polymers, 2023, doi:10.3390/polym15183712_

Round 1

Reviewer 1 Report

This manuscript reviews the potential and latest research related to Polydopamine as Molecularly Imprinted Polymers in the Sensing technology application. This manuscript is very easy to read and written well. Before this review manuscript can be publish, several minor concerns need to be amended and address. Here are some comments:

Concern:

  1. Self-citation: I would suggest the authors to cite research paper done by the authors (appox. 9-10 publications) to more significant sentence, explanation and scientific elaboration, and not only citing the example polydopamine usage in sensing applications (introduction section)
  2. In tables that have column for “comments”, kindly please explain in the paragraph about what the additional comments will provide to the reader so that the reader will appreciate the significance of that column.
  3. Missing Section 3 introduction as the numbering suddenly start at 3.1.
  4. Section 3.1: The authors explained dopamine has gained significant attention as a monomer in the synthesis of IIPda. However, only one work was cited for this section, indicating that the IIPda is not that favourable. In the section, authors only wrote about the advantages of IIPda. I would suggest to also write the drawbacks so that the readear can fully understand the fully potential of IIPda in current time and future.
  5. Section 6.2 and Section 6.3: I would like to suggest that to have a better presentatation this section (very important section for this review paper), research works done related for this section should be simplified in tabulated form and the method that they applied should be shown in figure form.

Reviewer 2 Report

Lamaoui et al, summarized the efforts of use of dopamine in molecular imprinting. Review covers important studies, however, needs substantial improvement before its publication. My detailed comments are below.

1.      Graphical abstract; Why epitope and protein are categorized separately? Epitopes are part of protein and with epitope imprinting also proteins have been measured.

2.      Line 47; monomer or as a functional monomer.

3.      Sec 2; How QCM crystal can be core material? It can provide surface to grow or deposit.

4.      Ref 32, results shown is original paper did not confirm any catalytic effect. There is no direct role of Fe2O3 in recognition. except easy separation.

5.      Author provided several references to confirm suitability of dopamine, however, these polymers are self-limiting and thickness of such polymer can be between 5-10 nm. Stability of molecular cavities in such thin polymer can be an issue. Author did not discuss this point.

6.      Additionally, in classical imprinting cross-linkers are added to provide 3D to molecular cavities. We do not see such dimensionality when dopamine is use as a monomer. This is an important issue.

7.      Why self-polymerization is more preferred over electropolymerization approach? Author should include this discussion to pass important information to its reader.

8.      Line 320, MIP based sensing system cannot be consider as a biosensor according to IUPAC definition.

9.      Morphology; Author should include some representative morphology figure.

10.   How wettability can improve recognition? Please add discussion on this point.

11.   Sec 6; lines 418-420; Author mentioned the MIP dopamine is ground breaking material with robustness, however, in conclusion it is clearly mentioned that such film is not much stable with time. Author should add own critical comments rather than repeating statement mentioned in original article.

12.   Author mentioned in conclusion that template extraction is difficult due to presence of covalent bond between template and dopamine during imprinting, however, did not support this statement with suitable references.

13.   line 611-613 and 594-595 is same statement.

.

Round 2

Reviewer 2 Report

Author response to my comments are satisfactory. Quality of review is improved. Now manuscript is suitable for publication.